**Subject Category:**
Biology (whole organism)

health and disease and epidemiology/ biomathematics

transmission model, *Campylobacter* transmission, campylobacteriosis, house flies, mechanical vectors, climate change

**Author for correspondence:**
Amy L. Greer
e-mail: agreer@uoguelph.ca

# Modelling the transmission dynamics of *Campylobacter* in Ontario, Canada, assuming house flies, *Musca domestica*, are a mechanical vector of disease transmission

Melanie Cousins[1,2], Jan M. Sargeant[1,2], David Fisman[3] and Amy L. Greer[1,2]

[1]Department of Population Medicine, Ontario Veterinary College, University of Guelph, Guelph, Ontario, Canada
[2]Centre for Public Health and Zoonoses, University of Guelph, Guelph, Ontario, Canada
[3]Department of Epidemiology, Dalla Lana School of Public Health, University of Toronto, Toronto, Ontario, Canada

MC, 0000-0003-3725-4058

*Campylobacter*'s complicated dynamics and multiple transmission routes have made it difficult to describe using a mathematical framework. Vector-borne disease transmission has been proposed as a potential transmission route of *Campylobacter* with house flies acting as a mechanical vector. This study aimed to (i) determine if a basic *SIR* compartment model that included flies as a mechanical vector and incorporated a seasonally forced environment compartment could be used to capture the observed disease dynamics in Ontario, Canada, and (ii) use this model to determine potential changes to campylobacteriosis incidence using predicted changes to fly population size and fly activity under multiple climate change scenarios. The model was fit to 1 year of data and validated against 8 and 12 years of data. It accurately captured the observed incidence. We then explored changes in human disease incidence under multiple climate change scenarios. When fly activity levels were at a 25% increase, our model predicted a 28.15% increase in incidence by 2050 using the medium−low emissions scenario and 30.20% increase using the high emissions scenario. This model demonstrates that the dynamics of *Campylobacter* transmission can be captured by a model that assumes that the primary transmission of the pathogen occurs via insect vectors.

# 1. Background

Campylobacteriosis (the infection caused by the bacteria *Campylobacter*) affected 21 in 100 000 people in Ontario in 2017, making it the most common gastrointestinal illness in Canada [1]. Humans can contract these bacteria from a wide variety of sources including: contaminated food and water, contact with animals and animal faeces and contact with an infected individual or their faecal matter [2–4]. In a study in Ontario, of domestic cases of known exposure, 63% were food-related, 27% from contact with animals, 6% from other individuals and 3% from contaminated water [5]. Also, Ravel *et al.* [6] researched campylobacteriosis source attribution through exposure assessment and comparative genomic fingerprinting using isolates from clinical cases and potential sources in Canada [7]. They found that chicken meat was the most common source (65–69%) followed by contact with cattle or cattle faeces (14–19%), and lastly meat from cattle was of minor importance [8]. This evidence suggests that contaminated food and contact with animals are of highest importance as the source of campylobacteriosis in Canada.

Human campylobacteriosis exhibits seasonal fluctuations in disease incidence with peaks in the summer months, June–August [8–12]. There are many hypotheses for the observed seasonality including: environmental/climactic changes, human behavioural changes and, more recently, flies as mechanical vectors and seasonal fluctuations in fly populations [9–11,13]. There is evidence that campylobacteriosis is associated with certain environmental and climatic factors such as increased temperatures, increased humidity and higher water flows [14–16]. As *Campylobacter* use the environment to move between human and animal hosts, it is important to know how these bacteria are affected by external factors.

It has also been noticed that the seasonal fluctuations in campylobacteriosis coincide with times of highest fly population density and activity level [13]. It has been demonstrated that flies carry bacteria including *Campylobacter*, and these bacteria can be transferred to food and surfaces from which humans can then become infected [17]. Flies can also encounter these bacteria, carry them on their bodies and transfer them among agricultural settings, such as between pens or barns, thus infecting food animals which can then be contacted or consumed by humans [13].

This theory requires attention, as fly population size and activity are subject to increase along with the number of flies surviving the winter season under predictions of climate change [18,19]. Climate change is predicted to cause an increase in temperature, humidity and precipitation in Canada and places of similar latitude [20]. It is also expected that the winter season will be shorter and warmer with more precipitation falling as rain instead of snow [21]. In a study in the UK, temperature, humidity and precipitation were all highly correlated with fly population size [18]. Using a model, these authors projected the annual size of house fly populations under medium–low and high carbon emission scenarios [18]. Their results suggest up to a 244% increase in fly population size by 2080 under a high emissions scenario over the population in 2003 [18]. Fly activity is also predicted to increase as temperatures rise [19]. Schou *et al.* [19] found that both sexes of house flies' daytime activity increased with temperature until a threshold of 30–35°C, respectively. Therefore, as the ambient temperature rises, flies may become more active throughout the day. Since house flies are very sensitive to changes in the environment, it is important to know how flies may react to climate change and in turn change the dynamics of campylobacteriosis.

A mathematical modelling framework can be used to simulate the spread of a pathogen through a population of individuals in order to quantify disease outcomes, including the burden of disease, the number of secondary cases arising from a single case ($R_0$) and/or the population attack rate [20]. Models can also be used to examine hypotheses related to data gaps, or disease prevention and intervention strategies [20].

Simple compartmental models have been expanded to capture the dynamics of waterborne diseases such as cholera [21–23]. In most cases, this was done by adding a water compartment into which infected individuals shed the pathogen. Susceptible humans can then become infected by contact with this infected water reservoir or by contact with an infected individual. This allows the model to capture traditional person–person transmission (aka the 'fast loop' [22]) as well as person–water–person transmission (or the 'slow loop' [22]). These types of models allow researchers to gain a more complete picture of the disease dynamics as well as test the impact of intervening on the different transmission routes.

Compartmental models have also been expanded to model vector-borne infectious diseases, known as Ross–MacDonald models [24–26]. Ross and MacDonald developed a model for mosquito-borne pathogen transmission that included latency due to the pathogen's life cycle [26,27]. In the Ross–MacDonald model structure, a susceptible vector obtains the pathogen from a host during blood

feeding. Once the pathogen has multiplied to a sufficient level in the vector, it can then be passed to a new susceptible host during a subsequent feeding [24,25]. These models have been adopted as the standard framework for many vector-borne diseases [26].

Owing to *Campylobacter*'s zoonotic disease dynamics and multiple transmission routes, it is a complicated host–pathogen system to model. Skelly and Weinstein modelled human campylobacteriosis that explicitly looked at infection from aquatic environments contaminated by human and animal faeces (treated and untreated water), and through food consumption, preparation and processing [27]. However, there are no models that attempt to capture the observed seasonality either through the environment, through fly dynamics or through these two sources in combination. The objectives of this study were to: (i) determine if a basic *SIR* compartment model that incorporates flies as a mechanical vector, and also incorporates a seasonally forced environmental reservoir compartment, could be used to capture the observed disease dynamics in Ontario, Canada, and (ii) use this model to explore possible changes to campylobacteriosis incidence using projected changes to fly population dynamics and fly activity level under different climate change scenarios.

# 2. Methods

## 2.1. Case data

We used two sets of data to parametrize and validate our model. Firstly, we had access to confirmed campylobacteriosis cases from Public Health Ontario (PHO) from 1 January 2005 to 31 December 2013. A positive individual had gastrointestinal illness symptoms and either the pathogen isolated from stool or body fluids, or had an epidemiological link to one or more laboratory-confirmed cases. The positive cases were reported to the integrated Public Health Information System (iPHIS). Cases that had travelled outside of Ontario within the incubation period of *Campylobacter* were excluded as it was assumed that these do not represent locally acquired cases. Over the 9-year period, there were a total of 27 956 confirmed cases in Ontario. The campylobacteriosis cases showed seasonality, with most cases occurring in the spring and summer months (June–September). The second form of data was publicly available 'Monthly Infectious Disease Surveillance Reports' provided by PHO which is a summation of cases from iPHIS on a provincial level.

## 2.2. Model structure

A deterministic *SEIR* model was developed that included the addition of an environmental reservoir ($B$). This reservoir is a placeholder to include many of the potential transmission routes: contaminated water, contaminated food, contact with animals and other environments contaminated by human and animal faeces. The environmental reservoir is seasonally forced to account for the changing levels of bacteria in the reservoir depending on the season. *Campylobacter* are sensitive to changes in the environment, which results in variability in the bacterial load in the environmental reservoir dependent on the season [8]. This dynamic occurs via direct changes to the biology of the bacteria in the environment as well as changes in the other aspects of the environmental reservoir. For example, high temperature and water levels are associated with increased odds of *Campylobacter*-positivity on farms [28]. This could lead to increased shedding of the pathogen and therefore an increased environmental load [8]. High temperatures are also associated with increased carcass levels of *Campylobacter* on poultry which could lead to increased risk of food contamination [10,29]. In order to mathematically allow for the seasonal oscillation of the environmental compartment ($B$), an 'environmental parameter' was added to the model ($\zeta$). This parameter is a scaling/augmenting factor (refer to appendix A, equation (A 5)).

This *SBEIR* model was then further expanded to include house flies, *Musca domestica*, as mechanical vectors (e.g. insects that carry the pathogen on the outside of the body and transmit through physical contact [30]). In this model (figure 1), a susceptible human ($S_h$) can become infected in three different ways: contact with an infectious human ($\beta_i$), contact with the environmental reservoir ($\beta_b$) or consuming food that has been contaminated by a 'contaminated' fly ($\beta_f$).

Susceptible flies enter the population at a seasonally fluctuating birth rate ($\mu_b$) and leave the population at a seasonal rate dependent on the number of flies in each compartment ($\mu_d$). The birth rate is seasonal because egg laying rates and larval development times fluctuate with temperature [13,31]. Female flies have more eggs in their lifetime at warmer temperatures (117.8 ± 36.5 at 20°C versus 494.9 ± 73.2 at 30°C) [32]. Also, warmer temperatures along with an increase in rainy days leads to larvae being able to

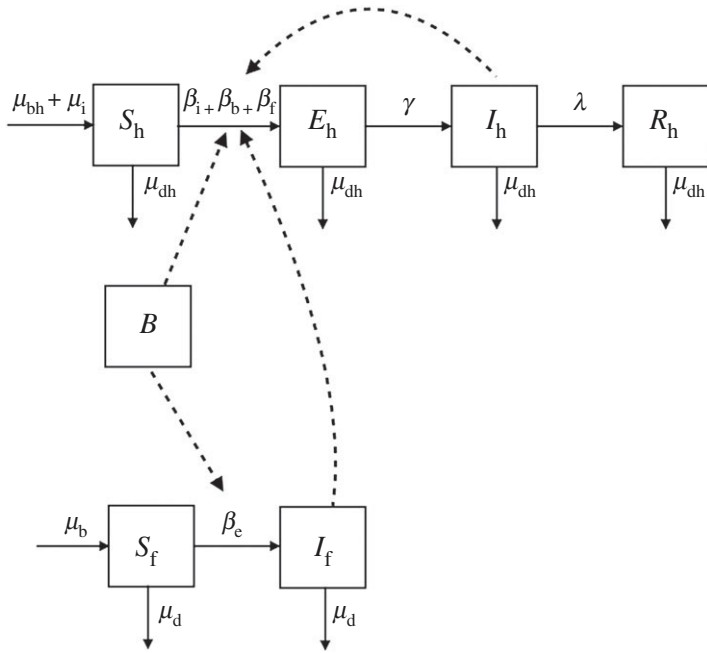

**Figure 1.** Compartmental diagram showing *Campylobacter* transmission in humans (top, h subscript) driven by flies (bottom, f subscript). Susceptible humans ($S_h$) are exposed ($E_h$) before becoming infectious to others ($I_h$) and then recover ($R_h$). Susceptible house flies ($S_f$) become contaminated ($I_f$) when they contact the environment ($B$). Person–person, environment–person and mechanical vector transmission due to flies are denoted by the dotted lines. Rates of change are represented by Greek letters above the arrows.

**Table 1.** Model parameters with values from the literature (ranges used for sensitivity analysis) and assumptions.

| symbol | definition | value (range) | reference |
|---|---|---|---|
| *human demographic parameters* | | | |
| $\mu_{bh}$ | human birth rate | 0.000026 days$^{-1}$ | [33] |
| $\mu_{dh}$ | human death rate | 0.000019 days$^{-1}$ | [33] |
| $\mu_i$ | human immigration rate | 0.000016 days$^{-1}$ | [33] |
| *transmission parameters* | | | |
| $\beta_i$ | person–person transmission rate | fitted | |
| $\beta_b$ | environment–person transmission rate | fitted | |
| $\beta_f$ | mechanical vector transmission rate | fitted | |
| $\beta_e$ | environment–fly transmission rate | fitted | |
| *disease parameters* | | | |
| $\gamma$ | latent period | 0.65 (0.33–1) days$^{-1}$ | [2,34] |
| $\lambda$ | duration of infection | 0.0476 (0.014–0.142) days$^{-1}$ | [2,35–37] |
| $\zeta$ | environmental parameter | fitted | |
| *fly demographic parameters* | | | |
| $\mu_b$ | fly birth rate | fitted | |
| $\mu_d$ | fly death rate | fitted | |

develop into adult flies within days compared to months (tables 1 and 2) [13]. These two factors in combination allow the fly population to increase exponentially as temperatures rise into the summer months. The fly birth rate includes both fly births from domestic flies that survived the winter and the influx of flies from warmer climates that occurs as temperatures rise [38]. Therefore, the birth rate is

**Table 2.** Model initial conditions with values from the literature and assumptions.

| symbol | definition | value | reference |
|---|---|---|---|
| $S_h 0$ | initial human susceptible population | fitted | |
| $B 0$ | initial environmental load | fitted | |
| $E_h 0$ | initial human exposed population | 0 | assumption |
| $I_h 0$ | initial human infectious population | 9 | PHO dataset |
| $R_h 0$ | initial human recovered population | fitted | |
| $S_f 0$ | initial fly susceptible population | fitted | |
| $I_f 0$ | initial fly infectious population | 0 | assumption |
| $w$ | number of flies survive winter | fitted (same as $S_f 0$) | |

independent of the number of flies in the population. The death rate is seasonal because temperature also affects adult fly survival [32]. Flies thrive at mean temperatures of 20–25°C. Survival decreases above and below this range [32].

In temperate climates, such as Ontario, most flies do not survive the cold winter temperatures [32]. In order to capture this, the model removes all flies above the initial fly population ($w = S_f 0$) at the end of each warm season. Therefore, the model restarts every year with the same number of susceptible flies and zero 'contaminated' flies. Susceptible flies become 'contaminated' when they contact the environmental reservoir ($\beta_b$). The flies pick up bacteria on their bristles. When they land on human food, they leave the bacteria behind. These bacteria can then infect a susceptible human when the food is consumed ($\beta_f$). It is assumed that the contaminated flies remain contaminated until they die. This dynamic disease transmission process is represented by equations (A 1)–(A 7) in appendix A.

## 2.3. Model assumptions

The model assumed homogeneous mixing between the entire human population of Ontario, Canada. The model assumed that humans acquire lifelong immunity to *Campylobacter* after they have recovered from an infection. Waning immunity was added to the model and had minimal effect on the model outcomes and therefore was removed for simplicity. Owing to the low case fatality rate [39], the model also assumed that there was no difference in death rates for those who had been infected and those who were uninfected.

Many of the assumptions revolve around flies and their contact. There are few empirical data on fly contact rates with both humans and the environment. Therefore, these parameters were estimated through model fitting.

## 2.4. Model fitting and validation

Owing to the number of unknown parameters, the model was fit to existing data in order to estimate these parameters. This was done by fitting the model's unknown parameters to the first year of the PHO dataset (1 January to 31 December 2005) using R's optimizing function. This is an optimization technique based on Nelder–Mead, quasi-Newton and conjugate-gradient algorithms for general purpose [40]. Initial parameter estimates were defined along with upper and lower bounds using the 'L-BFGS-B' method. Model fit was determined graphically by visually comparing the model output to the observed daily incidence data.

The model was validated using two techniques. First, the model was run for the remaining duration of the dataset (2006–2013). The predicted daily incidence was compared to the observed PHO incidence from 1 January 2006 until 31 December 2013. This appeared to be a good fit graphically and, therefore, a second validation step was done to ensure the model outputs were accurate. Using the model-predicted cumulative incidence from 1 January 2005 to 31 December 2017, the cumulative incidence was compared to incidence reported in the 'Monthly Infectious Disease Surveillance Report' by PHO, with 15% of the cases removed to account for the proportion of cases that were assumed to have acquired the bacteria through international travel [1,39].

**Table 3.** Parameters used to calculate changes in fly population size and fly activity levels under climate change scenarios.

| | $\mu_b$ | $\mu_d$ | $S_f0/w$ | $\beta_e$ | $\beta_f$ |
|---|---|---|---|---|---|
| **baseline** | $2.15 \times 10^{-3}$ | $1.67 \times 10^{-3}$ | 4910 | $5.10 \times 10^{-11}$ | $1.03 \times 10^{-11}$ |
| **increase in fly population size** | | | | | |
| *medium−low emissions* | | | | | |
| 45.7% | $2.196 \times 10^{-3}$ | $1.696 \times 10^{-3}$ | 5250 | $5.10 \times 10^{-11}$ | $1.03 \times 10^{-11}$ |
| 84.3% | $2.175 \times 10^{-3}$ | $1.715 \times 10^{-3}$ | 5500 | $5.10 \times 10^{-11}$ | $1.03 \times 10^{-11}$ |
| 156% | $2.204 \times 10^{-3}$ | $1.734 \times 10^{-3}$ | 6000 | $5.10 \times 10^{-11}$ | $1.03 \times 10^{-11}$ |
| *high emissions* | | | | | |
| 45.7% | $2.196 \times 10^{-3}$ | $1.696 \times 10^{-3}$ | 5250 | $5.10 \times 10^{-11}$ | $1.03 \times 10^{-11}$ |
| 128% | $2.197 \times 10^{-3}$ | $1.727 \times 10^{-3}$ | 5750 | $5.10 \times 10^{-11}$ | $1.03 \times 10^{-11}$ |
| 244% | $2.186 \times 10^{-3}$ | $1.756 \times 10^{-3}$ | 6250 | $5.10 \times 10^{-11}$ | $1.03 \times 10^{-11}$ |
| **increase in fly activity** | | | | | |
| 25% | $2.15 \times 10^{-3}$ | $1.67 \times 10^{-3}$ | 4910 | $6.37 \times 10^{-11}$ | $1.29 \times 10^{-11}$ |
| 50% | $2.15 \times 10^{-3}$ | $1.67 \times 10^{-3}$ | 4910 | $7.64 \times 10^{-11}$ | $1.54 \times 10^{-11}$ |
| 75% | $2.15 \times 10^{-3}$ | $1.67 \times 10^{-3}$ | 4910 | $8.92 \times 10^{-11}$ | $1.80 \times 10^{-11}$ |
| 100% | $2.15 \times 10^{-3}$ | $1.67 \times 10^{-3}$ | 4910 | $1.02 \times 10^{-10}$ | $2.06 \times 10^{-11}$ |

## 2.5. Sensitivity analysis

A Latin hypercube sensitivity analysis was performed on all parameters in the model with results depicted as partial rank correlation coefficients. A univariate sensitivity analysis was performed on the initial conditions that had uncertainty or that were estimated through model fitting.

## 2.6. Climate change conditions

As the global temperature rises, the population dynamics of flies, as with many other vectors, are sure to change [18,19]. In order to capture expected changes in fly population size, we used population predictions from Goulson *et al.* [18] under moderate and high carbon emission scenarios. Under a medium–low emission scenario, the authors predicted a 45.7% increase by 2020, an 84.3% increase by 2050 and a 156% increase by 2080 of annual fly population size compared to baseline population size estimates from 2003 [18]. Under a high emission scenario, the authors predicted a 45.7%, 128% and 244% increase in annual fly population size [18]. This dynamic was captured in the model by changing the birth and death rate of the flies as well as increasing the number of flies that survive over the winter (table 3). The amount of fly activity also increases as temperatures increase and, therefore, is likely to increase under the warming temperatures associated with climate change [19]. This is of concern because as flies become more active, they may have more contact with the contaminated reservoir where they can pick up pathogens on their bodies. Therefore, this increased activity could also lead to more flies landing on our foods. An increase in fly activity was modelled by increasing the amount the flies contact the environment ($\beta_e$) and the amount the flies contact human food ($\beta_f$). We modelled this as a 25–100% increase in fly activity (table 3). These scenarios were also examined in combination (fly population × fly activity level) to determine the expected observed increase in human incidence of disease in these different scenarios. All scenarios were compared to the first year of the baseline scenario (1 January–31 December 2005). This year was chosen for comparison as the predictions by Goulson *et al.* [18] were compared to 2003 fly populations and the surveillance data show relatively stable incidence over the given time period.

# 3. Results

## 3.1. Model fit

The optimization function in R finds unknown parameter values by finding parameters to minimize the difference between the observed data and the model output. After parametrization (parameters found in

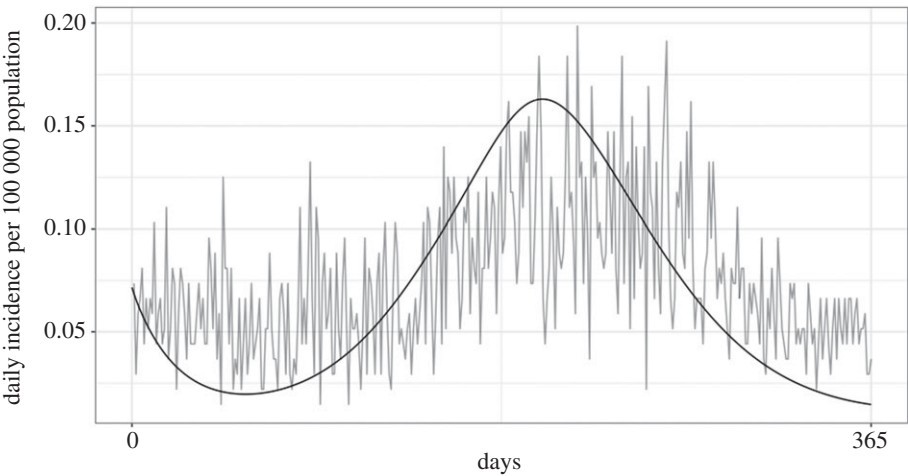

**Figure 2.** Model fit to PHO confirmed campylobacteriosis daily incidence from 1 January to 31 December 2005.

**Table 4.** Model parameters and initial conditions found through model fitting.

| symbol | definition | value |
|---|---|---|
| $\beta_i$ | person–person transmission rate | $1.034177 \times 10^{-13}$ |
| $\beta_b$ | environmental–person transmission rate | $3.380743 \times 10^{-07}$ |
| $\beta_f$ | fly–person transmission rate | $1.028844 \times 10^{-11}$ |
| $\mu_b$ | fly birth rate | $2.154924 \times 10^{-03}$ |
| $\mu_d$ | fly death rate | $1.67 \times 10^{-03}$ |
| $\beta_e$ | environment–fly transmission rate | $5.095899 \times 10^{-11}$ |
| Z | environmental parameter | $2.237077 \times 10^{-02}$ |
| $S_h 0$ | initial human susceptible population | $8 \times 10^{+06}$ |
| $B0$ | initial environmental load | $2.5 \times 10^{-02}$ |
| $R_h 0$ | initial human recovered population | $4.57 \times 10^{+06}$ |
| $S_f 0$ | initial fly susceptible population | $4.91 \times 10^{+03}$ |
| $w$ | number of flies survive winter | $4.91 \times 10^{+03}$ |

table 4), the model graphically appeared to have a good fit to the observed *Campylobacter* incidence in Ontario from 1 January to 31 December 2005 (figure 2), meaning that it found the minimal difference and best fit statistically. Since parameters were estimated using the first year of the data and appeared to have a good fit, the model was run for the next 8 years (the full duration of the dataset) using the best fit parameter values to validate the model. The daily incidence of the confirmed campylobacteriosis cases from PHO were graphically compared to the model and appeared to have a good fit (figure 3). For further validation, the cumulative incidence from 2005 to 2017 was compared to incidence reported in the 'Monthly Infectious Disease Surveillance Report' by PHO. This also appeared to have a good fit (figure 4).

## 3.2. Sensitivity analysis

From the Latin hypercube sensitivity analysis, the model was most sensitive to the latent period ($\lambda$), the environmental parameter ($\zeta$) and the fly death rate ($\mu_d$) as seen by the high partial rank correlation coefficients in figure 5. The model was moderately sensitive to the transmission parameters including flies and the environmental reservoir ($\beta_f$, $\beta_e$, $\beta_b$), but not to the person-to-person transmission rate ($\beta_i$).

From the univariate analysis, it was found that the model was least sensitive to the initial number of susceptible flies ($S_f 0$), and the upper bound of initial susceptible humans ($S_h 0$) (electronic supplementary material, table S1A). The model was highly sensitive to the initial environmental load ($B0$) and the lower

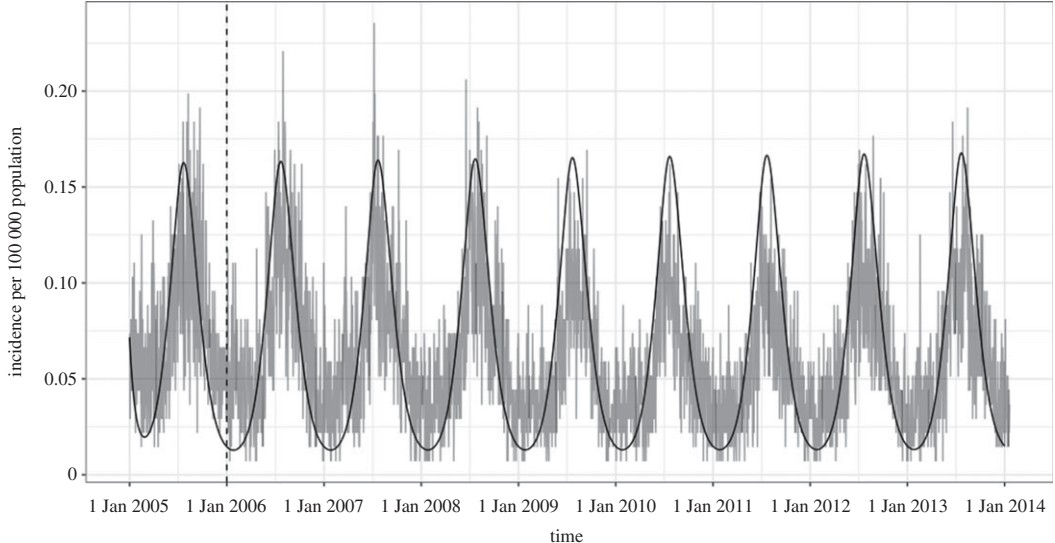

**Figure 3.** Model validation (black line) to daily incidence from PHO confirmed cases (grey line) from 1 January 2006 to 31 December 2013. The first year (before the dotted line) was used for model fitting.

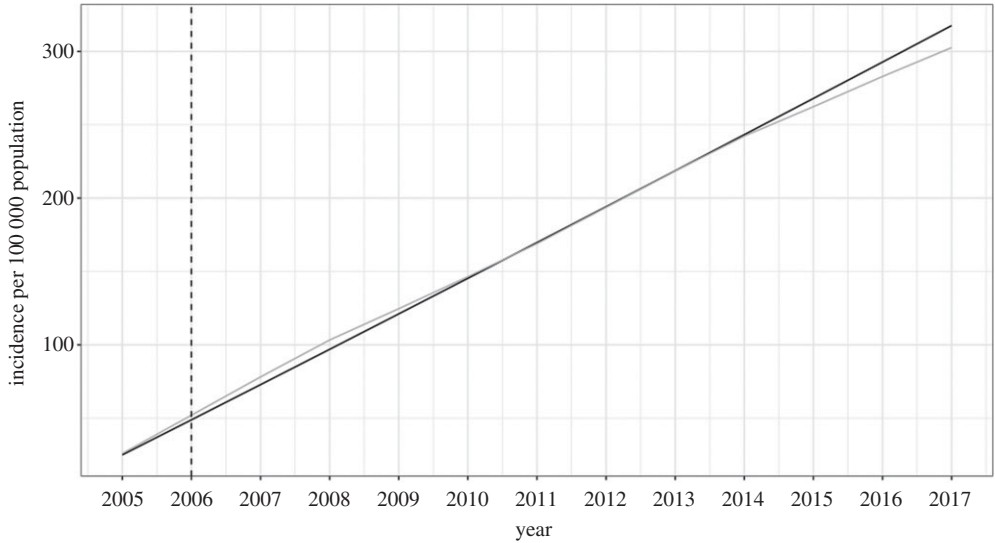

**Figure 4.** Model validation (black line) to cumulative incidence from PHO 'Monthly Infectious Disease Surveillance Reports' from 2005 to 2017 with 15% of cases removed due to international travel. The first year (before the dotted line) was used for model fitting.

bound of the initial susceptible humans ($S_h0$) (electronic supplementary material, table S1A). Therefore, it is necessary to have enough susceptible humans and enough bacterial load in the environment in order to initiate the spread of campylobacteriosis, until a certain threshold of susceptible humans is reached, in which case it can no longer spread to a greater extent. However, if there are more bacteria in the environment, this can lead to much greater outbreaks.

## 3.3. Climate change scenarios

Using the predictions of Goulson *et al.* [18], under medium−low emission scenarios, which correspond to a 156% increase in fly population size, the model showed that there could be a 6.67% increase in campylobacteriosis incidence by 2080 [18]. Under high emission scenarios which correspond to a 244% increase in fly population size, the model showed a 10.35% increase in incidence (figure 6; electronic supplementary material, table S2A).

When there was a 25% increase in fly activity, the model exhibited a 23.43% increase in human *Campylobacter* incidence. When fly activity was doubled, the model predicted a 93.73% increase in

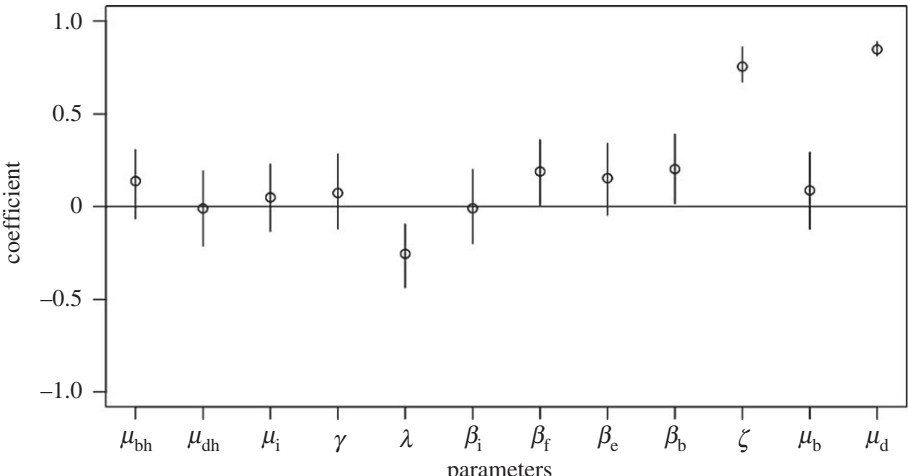

**Figure 5.** Results of Latin hypercube sensitivity analysis on all parameters in the model as partial rank correlation coefficients with the most sensitive parameters being the fly death rate ($\mu_d$) and the environmental parameter ($\zeta$).

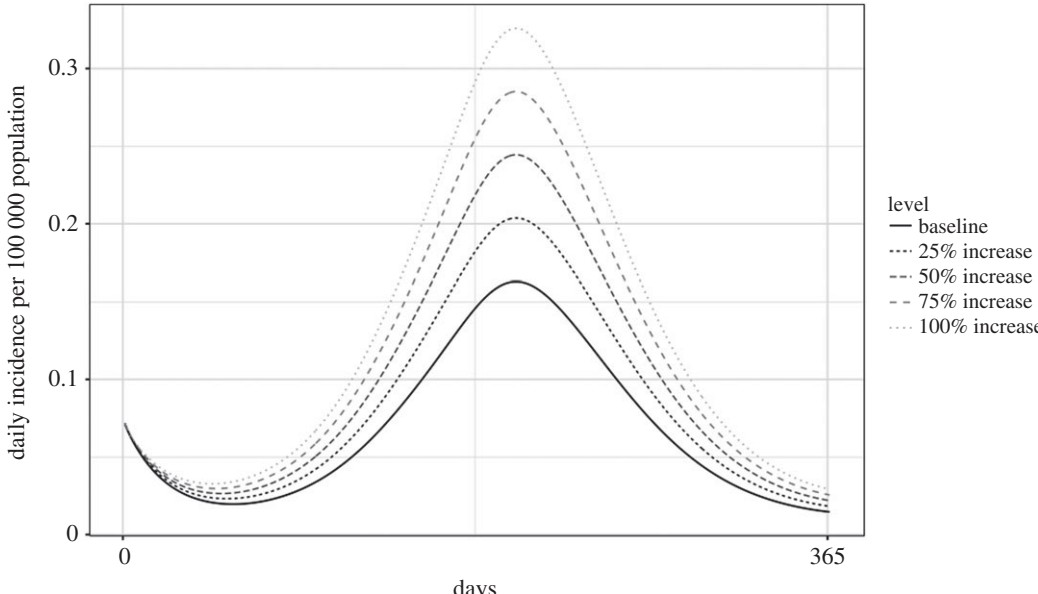

**Figure 6.** Predicted incidence of campylobacteriosis at varying levels of increase to fly population size.

incidence over the baseline of 25.04 cases per 100 000 population (figure 7; electronic supplementary material, table S2A).

More realistically, both phenomenon will occur under climate change scenarios. Therefore, we examined combinations in which all predicted population increases were run under the 25–100% activity increase scenarios. In this case, under medium–low carbon emissions, the model projected a 31.74% increase in incidence by 2080 if fly activity increased by 25% but up to a 107.02% increase if fly activity doubled. Under worst-case scenarios (high carbon emissions causing 244% population increase and 100% activity increase), the model projected a 114.43% increase in *Campylobacter* incidence in Ontario compared to the 2005 baseline (figure 8; electronic supplementary material, table S2A).

## 4. Discussion

Using a novel model structure, we have identified the environmental conditions that appear to describe the observed incidence of campylobacteriosis in Ontario. In addition, we have identified how the incidence of campylobacteriosis in the Ontario human population could change under different climate change

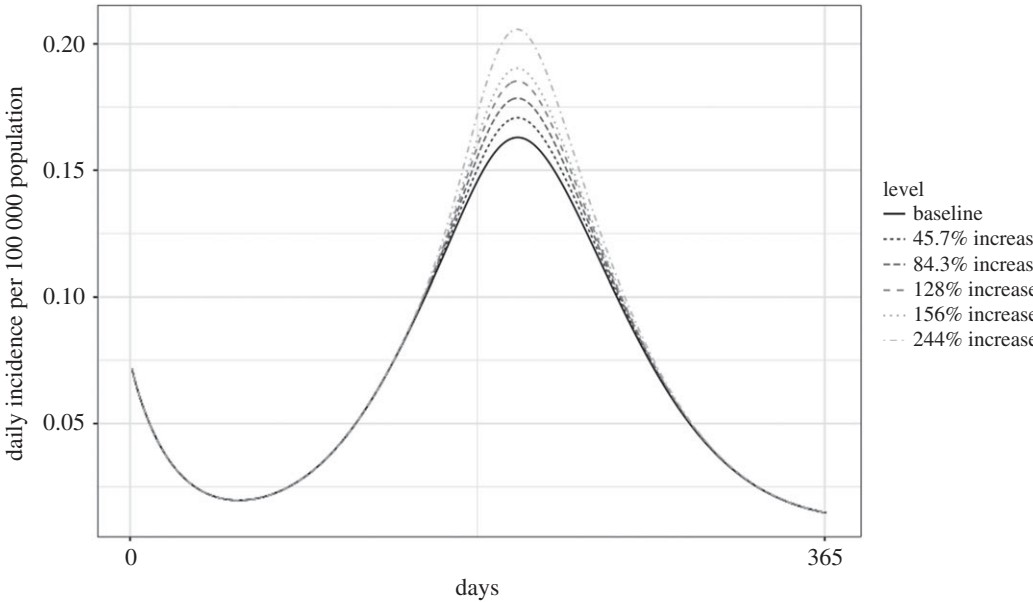

**Figure 7.** Predicted daily incidence of campylobacteriosis at varying levels of increase to fly activity.

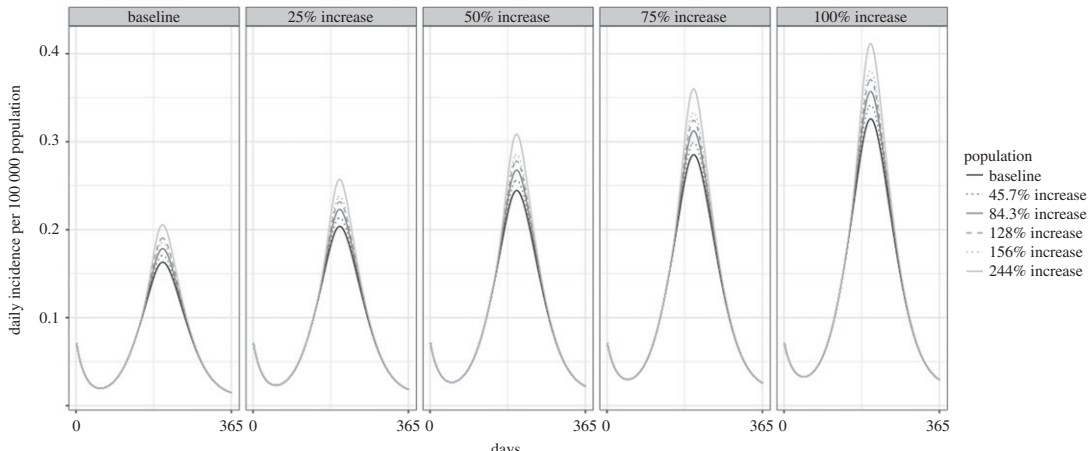

**Figure 8.** Predicted incidence of campylobacteriosis at a combination of varying levels of increase to fly population size (lines) and activity (panels).

scenarios that act to change fly populations and activity levels. This is an important step towards identifying evidence in support of this hypothesis which may lead to more research in this area.

## 4.1. Model results

According to our model results, increased fly activity is more influential in increasing the human incidence of campylobacteriosis compared to increased overall fly population size. For example, a 50% increase in fly activity resulted in a 46.9% increase in the incidence of campylobacteriosis; however, a 45.6% increase in fly population size resulted in a 1.9% increase in *Campylobacter* incidence. In the worst-case scenario, our fly population model predicted a 10.3% increase in disease incidence. This pattern has been noticed in other mathematical models of vector-borne diseases. For example, in a model of mosquito transmission in Africa that included seasonality, the authors found that insecticide-treated nets were more effective at controlling the spread of disease when compared with indoor residual spraying [41]. In this case, the nets are controlling the ability of mosquitoes to enter the homes and therefore are limiting contact with humans, whereas the spraying is controlling the size of the mosquito population. This has major implications for public health and, therefore, gives us insight into potential areas for intervention and control. For example, common practice for fly control includes spraying to reduce fly population size.

However, according to our model, small populations that are highly active still contribute significantly to transmission. Controlling fly activity or decreasing the transmission rate between flies and human food may be more beneficial in this case. Intervention studies have been performed in which fly screens were used to prevent the entrance of flies into poultry barns [42]. For example, there was a reduction in *C. jejuni* prevalence in poultry barns in Denmark from 41 to 10% in those that had fly screens [42]. These interventions appear to be successful at the flock level and may provide insight into future intervention strategies.

## 4.2. Limitations

Modelling *Campylobacter* transmission is difficult because of its complex dynamics. As a result, this model makes a number of assumptions and simplifications. For example, the homogeneous mixing assumption assumes that every individual has the same probability of contact with the environment, each other and the flies [43]. In reality, this may not be the case. For example, with many enteric diseases, there are high rates of transmission within households, but little transmission between other infected individuals [44]. Certain individuals may also have more contact with the environment or certain components of the environmental reservoir or may be more at risk for contact with flies, such as those living in rural areas or on farms. This model may under- or overestimate the incidence depending on the importance of the heterogeneity of the population. Therefore, other model structures may be required to overcome this assumption such as models stratified by age, living conditions or level of risk.

This model was calibrated and validated using human incidence only and therefore does not use independent data for the fly or environment reservoirs. Since empirical data for these reservoirs do not exist at this scale, these values were obtained through model fitting and could not be independently validated. This is an important limitation of the model and suggests that the collection of additional data for further model validation would be a useful next step.

Our model-predicted incidence aligns well with the years of observed data but does diverge after 2013 and predicts higher incidence for the subsequent 4 years. This may be an indication that our model may overestimate the burden of campylobacteriosis when run further into the future.

This model was created to explore if flies as a mechanical vector for disease transmission to humans was a viable hypothesis. There were no data on fly population dynamics and contact rates with both humans and the environment in Ontario, and therefore, these parameters were estimated through model fitting. The model was also sensitive to the transmission rates involving flies ($\beta_f$ and $\beta_e$). These parameters are influential because they determine the rate at which susceptible humans are becoming infected and, therefore, are big drivers of the disease dynamics. These parameters were also found through parametrization. Therefore, this model would benefit from further research into collecting empirical data to obtain more informative upper and lower bounds on these parameters to create more accurate and informative models.

Our results showed that increasing the amount of fly activity leads to a greater increase in incidence and, therefore, controlling fly contact may be a superior method of prevention than controlling fly population size. However, this may be a combination of the way in which we modelled increases to fly activity and the uncertainty around the amount fly activity will increase in the future. Increasing fly activity in our model involved increasing both the transmission parameter with humans ($\beta_f$) and the environment ($\beta_e$) by the intended percentage increase (i.e. a 25% increase in activity resulted in a 25% increase in $\beta_f$ and a 25% increase in $\beta_e$). This could have, however, been modelled as a synergistic effect by increasing each transmission parameter by 12.5% to create a total increase of 25%. Further research in this area is warranted.

An aspect of the biology that we did not address is that the retention of *Campylobacter* on flies may decline as temperatures increase [45]. This could be an important factor in how far flies can carry *Campylobacter* depending on the temperature. This could be tested in the future using our model in conjunction with projected fly population and activity changes under the different climate change scenarios.

Our model used predictions from Goulson *et al.* on climate change's effect on fly population size based on the assumption from the UK Climate Impacts Programme that the global temperature will increase by 2.34°C by 2080 in an optimistic medium–low carbon emission scenario and by 3.88°C in a high emission scenario [18,46]. Should the magnitude of climate change vary in North America, it would be expected that fly populations and activity would also vary accordingly.

# 5. Conclusion

A mechanistic infectious disease model for the transmission of *Campylobacter* in the Ontario human population in which flies act as a mechanical vector between contaminated environments and human food consumption was created. The model was able to capture the observed daily and cumulative incidence data, thus supporting the fly transmission hypothesis. Creating a model for *Campylobacter* which includes a seasonally fluctuating environmental compartment and fly populations will allow future researchers to test many different aspects of the transmission chain. This could include expanding the model to explicitly model specific transmission routes as well as test different prevention and control strategies.

Ethics. This project was approved by the University of Guelph, Research Ethics Board (REB#15NV011).

Data accessibility. Our data are deposited at GitHub: https://github.com/mmcousins/RoyalSocietyOpenScienceCousinsetal 2018.git.

Authors' contributions. All authors were involved in the study conception and design, analysis and interpretation of data, and drafting of the manuscript. M.C. built the model, analysed and interpreted the results and drafted the original manuscript. A.L.G., D.F. and J.M.S. assisted with data acquisition. All authors contributed to the revision of the manuscript for important intellectual content. All authors read and approved the final manuscript.

Competing interests. The authors declare that they have no competing interests.

Funding. This work was funded by the Canadian Institute of Health Research (CIHR) and Canadian Research Chairs Program (CRC). M.C. was supported by an Ontario Veterinary College scholarship.

Acknowledgements. The authors wish to thank Ms Enise Decaluwe-Tulk for her work to develop our teams' One Health database for enteric pathogens in the province of Ontario and for compiling all of the environmental data required for this project. The authors also wish to thank Public Health Ontario for access to the necessary data.

# Appendix A

Equations:

$$\frac{dS_h}{dt} = \mu_{bh}(S_h + E_h + I_h + R_h) + \mu_i(S_h + E_h + I_h + R_h) - \beta_i S_h I_f - \beta_b S_h B - \beta_f S_h I_f - \mu_{dh} S_h, \tag{A1}$$

$$\frac{dE_h}{dt} = \beta_i S_h I_h + \beta_b S_h B + \beta_f S_h I_f - \gamma E_h - \mu_{dh} E_h, \tag{A2}$$

$$\frac{dI_h}{dt} = \gamma E_h - \lambda I_h - \mu_{dh} I_h, \tag{A3}$$

$$\frac{dR_h}{dt} = \lambda I_h - \mu_{dh} R_h, \tag{A4}$$

$$\frac{dB}{dt} = \zeta\left(\sin\left(\frac{2\pi t}{365}\right)\right)B, \tag{A5}$$

$$\frac{dS_f}{dt} = \mu_b\left(-200\sin\left(\frac{2\pi t}{365}\right)\right) - \beta_e S_h B - \mu_d\left(-200\sin\left(\frac{2\pi t}{365}\right)\right)S_f, \tag{A6}$$

$$\text{and} \quad \frac{dI_f}{dt} = \beta_e S_h B - \mu_d\left(-200\sin\left(\frac{2\pi t}{365}\right)\right)I_f. \tag{A7}$$

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
