## [Reviewer comments · Royal Society Open Science]

Review History

RSOS-181394.R0 (Original submission)

Review form: Reviewer 1

Is the manuscript scientifically sound in its present form?

Yes

Are the interpretations and conclusions justified by the results?

Yes

Is the language acceptable?

Yes

Is it clear how to access all supporting data?

No

Do you have any ethical concerns with this paper?

No

Have you any concerns about statistical analyses in this paper?

I do not feel qualified to assess the statistics

Recommendation?

Accept with minor revision (please list in comments)

Comments to the Author(s)

This study aimed to determine if a basic SIR compartment model that included flies as a mechanical vector and incorporated a seasonally forced environment compartment could be used to capture the observed disease dynamics of *Campylobacter* in Ontario, Canada.

The model was calibrated using one season's public health surveillance data on campylobacteriosis incidence, and validated using multi-season data on campylobacteriosis incidence. The validated model was used to determine potential changes to campylobacteriosis incidence using predicted changes to fly population size and fly activity under multiple climate change scenarios.

This is a very careful mathematical modeling study. The analysis convincingly supports the conclusions.

The reviewer would like to point out one limitation on the modeling study, regarding model validation. The model, as described on page 25, has three kinds of hosts, human, flies, and environment (B). The surveillance data used for fitting and validation is only for human hosts; there is no independent data for flies or the environment to validate these two parts of the model. While it is reality that no such data exists, it is a limitation on the modeling study. The reviewer would like to see this limitation discussed in the discussion, so that other readers can be aware of it.

Decision letter (RSOS-181394.R0)

04-Jan-2019

Dear Professor Cousins

On behalf of the Editors, I am pleased to inform you that your Manuscript RSOS-181394 entitled "Modelling the transmission dynamics of *Campylobacter* in Ontario, Canada assuming house flies, *Musca domestica*, are a mechanical vector of disease tran" has been accepted for publication in Royal Society Open Science subject to minor revision in accordance with the referee suggestions. Please find the referees' comments at the end of this email.

The reviewers and handling editors have recommended publication, but also suggest some minor revisions to your manuscript. Therefore, I invite you to respond to the comments and revise your manuscript.

- Ethics statement

If your study uses humans or animals please include details of the ethical approval received, including the name of the committee that granted approval. For human studies please also detail

whether informed consent was obtained. For field studies on animals please include details of all permissions, licences and/or approvals granted to carry out the fieldwork.

- Data accessibility

If you wish to submit your supporting data or code to Dryad (<http://datadryad.org/>), or modify your current submission to dryad, please use the following link:
<http://datadryad.org/submit?journalID=RSOS&manu=RSOS-181394>

- Competing interests

- Authors' contributions

- Acknowledgements

- Funding statement

Because the schedule for publication is very tight, it is a condition of publication that you submit the revised version of your manuscript before 13-Jan-2019. Please note that the revision deadline

will expire at 00.00am on this date. If you do not think you will be able to meet this date please let me know immediately.

If your manuscript is newly submitted and subsequently accepted for publication, you will be asked to pay the article processing charge, unless you request a waiver and this is approved by

Royal Society Publishing. You can find out more about the charges at <http://rsos.royalsocietypublishing.org/page/charges>. Should you have any queries, please contact openscience@royalsociety.org.

on behalf of Dr John Dalton (Associate Editor) and Kevin Padian (Subject Editor)
openscience@royalsociety.org

Associate Editor Comments to Author (Dr John Dalton):

Associate Editor: 1

Comments to the Author:

Your paper has been accepted. Please see minor suggestions by our reviewer to improve your manuscript.

Editor comments to author:

Sorry for the delay; we got only one reviewer and then a second, who is three months late, and inasmuch as the first reviewer suggests only minor revisions we do not want to hold you up longer. Thanks for your patience and we look forward to your revision.

Reviewer comments to Author:

Reviewer: 1

Comments to the Author(s)

This study aimed to determine if a basic SIR compartment model that included flies as a mechanical vector and incorporated a seasonally forced environment compartment could be used to capture the observed disease dynamics of *Campylobacter* in Ontario, Canada.

The model was calibrated using one season's public health surveillance data on campylobacteriosis incidence, and validated using multi-season data on campylobacteriosis incidence. The validated model was used to determine potential changes to campylobacteriosis incidence using predicted changes to fly population size and fly activity under multiple climate change scenarios.

This is a very careful mathematical modeling study. The analysis convincingly supports the conclusions.

The reviewer would like to point out one limitation on the modeling study, regarding model validation. The model, as described on page 25, has three kinds of hosts, human, flies, and environment (B). The surveillance data used for fitting and validation is only for human hosts; there is no independent data for flies or the environment to validate these two parts of the model. While it is reality that no such data exists, it is a limitation on the modeling study. The reviewer would like to see this limitation discussed in the discussion, so that other readers can be aware of it.

Author's Response to Decision Letter for (RSOS-181394.R0)

See Appendix A.

Decision letter (RSOS-181394.R1)

14-Jan-2019

Dear Professor Cousins,

I am pleased to inform you that your manuscript entitled "Modelling the transmission dynamics of *Campylobacter* in Ontario, Canada assuming *Musca domestica* are a mechanical vector of disease transmission." is now accepted for publication in Royal Society Open Science.

on behalf of Dr John Dalton (Associate Editor) and Professor Kevin Padian (Subject Editor)
openscience@royalsociety.org

Appendix A

Reviewer 1:

1. *This study aimed to determine if a basic SIR compartment model that included flies as a mechanical vector and incorporated a seasonally forced environment compartment could be used to capture the observed disease dynamics of Campylobacter in Ontario, Canada.*

We appreciate that the aim of the study was clear.

2. *The model was calibrated using one season's public health surveillance data on campylobacteriosis incidence, and validated using multi-season data on campylobacteriosis incidence. The validated model was used to determine potential changes to campylobacteriosis incidence using predicted changes to fly population size and fly activity under multiple climate change scenarios.*

We appreciate that the methods of model calibration and validation were clear.

3. *This is a very careful mathematical modeling study. The analysis convincingly supports the conclusions.*

We gratefully thank the Reviewer for the review of this paper and are pleased with the feedback.

4. *The reviewer would like to point out one limitation on the modeling study, regarding model validation. The model, as described on page 25, has three kinds of hosts, human, flies, and environment (B). The surveillance data used for fitting and validation is only for human hosts; there is no independent data for flies or the environment to validate these two parts of the model. While it is reality that no such data exists, it is a limitation on the modeling study. The reviewer would like to see this limitation discussed in the discussion, so that other readers can be aware of it.*

Following the Reviewer's suggestion, we have added a section to the limitation section: This model was calibrated and validated using human incidence only and therefore does not use independent data for the fly or environment reservoirs. Since empirical data for these reservoirs does not exist at this scale, these values were obtained

through model fitting and could not be independently validated. This is an important limitation of the model and suggests that the collection of additional data for further model validation would be a useful next step. (Page 16, Line 356-360)